# Pregnancy Outcomes in Late Onset Pompe Disease

**DOI:** 10.3390/life10090194

**Published:** 2020-09-11

**Authors:** Ozlem Goker-Alpan, Vellore G. Kasturi, Maninder K. Sohi, Renuka P. Limgala, Stephanie L. Austin, Tabitha Jennelle, Maryam Banikazemi, Priya S. Kishnani

**Affiliations:** 1Lysosomal and Rare Disorders Research and Treatment Center (LDRTC), 3702 Pender Dr, Ste 170, Fairfax, VA 22030, USA; msohi@ldrtc.org (M.K.S.); rlimgala@ldrtc.org (R.P.L.); ttaber@ldrtc.org (T.J.); 2Division of Medical Genetics, Duke University Medical Center, 905 S. LaSalle Street, 4th Floor, Durham, NC 27710, USA; vellore.kasturi@duke.edu (V.G.K.); stephanie.austin@duke.edu (S.L.A.); priya.kishnani@duke.edu (P.S.K.); 3Lysosomal Storage Disorders Center, New York Medical College, 503 Grasslands Road, Ste 200, Valhalla, NY 10591, USA; Maryam_banikazemi@nymc.edu

**Keywords:** lysosomal disorders, glycogen storage disease type II, pompe disease, LOPD, pregnancy, enzyme replacement therapy

## Abstract

There is limited data on pregnancy outcomes in Pompe Disease (PD) resulting from deficiency of the lysosomal enzyme acid alpha-glucosidase. Late-onset PD is characterized by progressive proximal muscle weakness and decline of respiratory function secondary to the involvement of the respiratory muscles. In a cohort of twenty-five females, the effects of both PD on the course of pregnancy and the effects of pregnancy on PD were investigated. Reproductive history, course of pregnancy, use of Enzyme replacement therapy (ERT), PD symptoms, and outcomes of each pregnancy were obtained through a questionnaire. Among 20 subjects that reported one or more pregnancies, one subject conceived while on ERT and continued therapy through two normal pregnancies with worsening of weakness during pregnancy and improvement postpartum. While fertility was not affected, pregnancy may worsen symptoms, or cause initial symptoms to arise. Complications with pregnancy or birth were not higher, except for an increase in the rate of stillbirths (3.8% compared to the national average of 0.2–0.7%). Given small sample size and possible bias of respondents being only women who have been pregnant, further data may be needed to better analyze the effects of pregnancy on PD, and the effects of ERT on pregnancy outcomes.

## 1. Introduction

Pompe disease (PD; glycogen storage disease type II) (OMIM: 232300) is a rare lysosomal storage disorder resulting from the deficiency of the enzyme acid alpha-glucosidase (EC 3.2.1.20). Similar to many lysosomal storage diseases, the phenotypic spectrum in Pompe disease is varied, ranging from infantile, to juvenile and late-onset presentations. In the classic infantile form, patients develop cardiomyopathy, resulting in premature death. In late-onset Pompe disease (LOPD), while there is lack of significant cardiac involvement, proximal muscle weakness similar to that of limb-girdle muscular dystrophy causes progressive weakness.

LOPD can present at any age, and primarily lower limb muscles, paraspinal muscles, and respiratory muscles are affected. The diaphragm is often affected early in the disease course. The course is usually variable, but diaphragmatic weakness may occasionally precede peripheral muscle weakness. Respiratory complications are often the cause of significant morbidity and mortality [1]. Similar to other lysosomal storage disorders, genetic, epigenetic, and environmental factor including race, ethnicity, gender, and polymorphisms in other genes (angiotensin-converting enzyme (ACE), alpha-actinin-3 (ACTN3)) have been suggested to affect the onset and progression of Pompe disease [2]. In general, events acting as metabolic stressors, such as pregnancy are implicated as disease modifiers. For example, in a much-studied lysosomal storage disorder, Gaucher disease, pregnancy increases the risk of acute bone crises, even in otherwise asymptomatic patients [3]. However, similar studies regarding the impact of pregnancy are needed for LOPD. There is only limited information available as guidelines for clinicians in counseling women with Pompe disease who are planning to become pregnant. Currently, data used to counsel female patients with Pompe disease are derived from other muscular dystrophies, because neither obstetric risk, nor complications of pregnancy have been studied or reported in depth in LOPD. It has been shown that in other muscular dystrophies, such as fascioscapulohumeral dystrophy (FSHD), the rates for low birth weight and total operative deliveries were statistically higher than the national rates in the general population. In addition, worsening of muscle weakness was observed in 24% of gestations and did not usually resolve after delivery [4,5].

The course of Pompe disease has been significantly altered with the advent of enzyme replacement therapy (ERT). The mannose targeted recombinant human alpha glucosidase improves cardiac and skeletal muscle involvement [6,7]. Better outcomes are usually achieved when ERT is started early, prior to progressive muscle damage [8]. However, the skeletal muscle response has been more variable than cardiac muscle. In LOPD, ERT was shown to improve muscle strength with improved 6-min walk test results [9]. However, often with a delayed diagnosis, and most variable presentation, the data derived from clinical studies may not reflect the long term and true disease outcomes. Thus, the information derived from the natural history of patients treated with ERT will be vital for clinicians to decide when to start therapy and how to follow treated patients with LOPD. ERT has been available commercially for the past decade, and alglucosidase alfa (Lumizyme^®^) was approved by the Food and Drug Administration (FDA) in 2010 for the treatment of patients with LOPD. It has been initially classified by FDA as Pregnancy Category B but later changed to Pregnancy Category C, according to which potential benefits may warrant use of alglucosidase alfa in pregnant women despite potential risks. There were no studies of alglucosidase alfa in pregnant women during the drug development phase. In animal reproduction studies, no effects on embryo-fetal development were observed in mice or rabbits given daily administration of alglucosidase alfa at the recommended human bi-weekly dose during the period of organogenesis. An increase in mortality was observed in pups when alglucosidase alfa was administered every other day in mice during the period of organogenesis through lactation at the recommended human bi-weekly dose [10].

While there are few sporadic case reports of pregnancy outcomes in Pompe disease patients who have continued ERT through the pregnancy with some complications, there is no extensive data available about the use of alglucosidase alfa during pregnancy or the effects on pregnancy outcomes [11,12,13,14,15,16]. This study specifically focuses on female patients with LOPD, to characterize the obstetric history and concomitant problems in females with Pompe disease, exploring the disease progression in patients who have been pregnant, including the impact of ERT on the outcomes of pregnancy and the fetus. The results are expected to help clinicians in counseling and caring for women with Pompe disease, who are planning to become pregnant, and during the pregnancy.

## 2. Results

The demographics and clinical characteristics of patients with LOPD, including diagnostic modality, presentation, and disease progression in relationship to the pregnancy history are summarized in Table 1. Among 25 female patients with Pompe disease, the mean age of disease onset was 33.8 years (range 7–66 years), and the mean age at diagnosis was 42.1 years (range 24–60 years). The most common diagnostic method was muscle biopsy (20/25), followed by molecular testing (17/25). Only three patients were diagnosed through noninvasive methods and one patient underwent a liver biopsy. In the majority of patients (21/23), the diagnosis of Pompe disease was made after the completion of all pregnancies. The presentation of Pompe disease was variable. In addition to the most common complaint of proximal muscle weakness with the inability to climb stairs or rise from a squatting position, some presented with nonspecific symptoms such as fatigue and flu-like symptoms. Three patients presented with shortness of breath. The current state of disease progression was ascertained through the use of assistive ambulation. At the time when the surveys were completed, more than half of the patients (14/25) reported the use of assistive devices for ambulation, and one patient was wheelchair bound. Only three patients reported respiratory compromise either with decreased exercise capacity or shortness of breath, and none of the patients received respiratory support, neither invasive nor noninvasive ventilation.

Among the 25 subjects, 3 reported infertility, and assisted reproductive technology was utilized in 1 patient recently diagnosed with Pompe disease, who elected cryopreservation of embryos prior to commencing ERT. There were 7 (12.07%) spontaneous first trimester miscarriages, 50 (86.2%) live births (including one pair of twins), and 2 (3.4%) stillbirths. Of the 52 births, the majority (80.8%) were vaginal deliveries and 10 (19.2%) were Cesarean sections. The preterm birth rate was 13.46% (7/52) and low birth weight rate was 7.7% (4/52). One patient had prolonged recovery from general anesthesia as a complication. Only one infant was born with a congenital defect, multi-cystic kidneys, and the mother was not exposed to ERT before or during pregnancy. The majority of patients (20/23) had pregnancies before being diagnosed with Pompe disease, and thus were not on ERT. Of those 20 patients, a total of 42 pregnancies were reported. Five of those patients reported postpartum fatigue and one patient was diagnosed with Pompe disease at the time of her pregnancy. During pregnancy, 38.5% of the subjects reported back pain. One patient reported improvement of symptoms during the course of her pregnancy.

Among those 23 subjects who reported one or more pregnancies, two subjects conceived while on ERT and continued therapy through pregnancies with worsening of weakness during pregnancy and improvement postpartum. One patient was started on ERT during pregnancy. For all the patients who were on ERT during the pregnancy, there were no adverse events during the administration of ERT (Table 2).

Back pain, which was noted as one of the presenting symptoms for Pompe disease, was also the most common Pompe disease related symptom during pregnancy and 84% of the subjects reported back pain sometime during their pregnancy. One patient on ERT had symptoms related to Pompe disease worsen during pregnancy. One patient had symptoms improve during pregnancy while three patients had symptoms worsen postpartum. One patient on ERT had symptoms improve postpartum.

Complication rates in pregnancy or childbirth were not higher in study subjects, compared to the CDC’s National Vital Statistics Report except for an increase in the rate of stillbirths (3.8% for study subjects compared to 0.2–0.7% for the national average). When compared with data from the CDC, the mode of delivery, and rate of preterm delivery were not different among females with Pompe disease while the anesthesia complications were higher in incidence (5% for study subjects compared to 0.1% for the national average) (Figure 1).

For the pregnancy and fetal outcomes, all but two pregnancies resulted in a healthy newborn. There was one full term unexplained still birth and one still birth at 24 weeks gestation secondary to a staphylococcal infection. One fetus had a right multi-cystic kidney which spontaneously resolved (Figure 1).

## 3. Discussion

Gender and sex may have an effect on pathophysiology of many disease states, such as autoimmune, cardiovascular, and neurological disorders, but may also affect therapeutic outcomes by having an influence on the distribution and efficacy of pharmaceuticals. In addition, gender-related health behavior and choices may overall impact disease progression and outcomes [17]. In this report, we assessed a cohort of female patients with LOPD recruited from two different centers in the USA. This cohort includes patients both treatment naïve or on ERT, and those who have been pregnant and/or are contemplating pregnancy.

It is now established that the initial diagnosis of LOPD is significantly delayed, often decades after the onset of first symptoms. The suggested reasons for this delay are multiple and some include the indeterminate initial presentation other than muscle weakness. When muscle weakness ensues, there is an overlap of findings with other muscular dystrophies such as the Limb Girdle Muscular Dystrophy group.

In this cohort, the majority of patients (20/25) had already been pregnant or had completed the planning of their families, at the time of the diagnosis of Pompe disease. This finding has been replicated in other reported individual cases, and in two different studies from Europe [18,19]. In a study conducted in the Netherlands, among 94 patients with LOPD, muscle strength deteriorated about −1.3% points/year for manual muscle testing and of −2.6% points/year for hand-held dynamometry; both p < 0.001. Prolonged disease duration (>15 years) and pulmonary involvement predicted a faster decline, while a consistent decline in pulmonary function was consistent in all patients [19]. Since only a minority of patients (3/25) reported significant pulmonary decline, we have ascertained disease progression through the use of assistive mobility devices. More than half of patients (14/25) were using an assistive device, such as canes, walkers, scooters, and wheelchairs, for mobility at the time of the study. Three patients reported worsening of symptoms in relationship to mobility, specifically during pregnancy. While our study is not intended to assess disease progression individually, a significant number of patients in this cohort reported obvious disease progression even prior to the diagnosis, consistent with the diagnostic lag in other reports.

One could argue that females have more frequent access to health services during reproductive ages. Although these patients were followed closely during their pregnancies, only one was diagnosed at the time of her pregnancy. Thus, the relative inexperience of medical professionals with the different presentations of LOPD could be added to the above reasons for diagnostic delays observed in LOPD. Currently, the utilization of expanded Next Generation Sequencing (NGS) panels, and Whole Exome Sequencing are suggested as the first tier diagnostic testing in patients presenting with atypical symptoms or in those suspected to have a muscle disease of undetermined etiology to avoid such diagnostic delays and improve treatment outcomes [20].

Among muscular dystrophies, and other neuromuscular disorders, only Myotonic Dystrophy is known to affect fertility, and in females, menstrual irregularities are often reported. While this study demonstrates that fertility does not appear to be affected in Pompe disease, Pompe disease could have contributed to future reproductive choices in this patient cohort [21]. The physiological changes associated with pregnancy do not directly affect skeletal muscle; however other alterations including change in cardiac output with increased heart rate and stroke volume, alterations in ligaments, joints and smooth muscle may impact a patient with Pompe disease [21]. As demonstrated in this cohort, the impact of pregnancy on Pompe disease can be varied, while in the majority of patients it may not have an immediate clinical impact. Pregnancy may initiate symptoms of Pompe disease, leading to subsequent investigations and diagnosis, and in some patients, pregnancy may worsen symptoms in Pompe disease. The influence of pregnancy on respiratory function could be particularly of importance in a female patient with LOPD who already has compromised respiratory function [22]. Changes in respiratory parameters, such as Peak Expiratory Flow should be interpreted cautiously in pregnant women with impaired lung function, and should not be mistaken for an improvement in function [23]. In our series, only three patients reported respiratory compromise either with decreased exercise capacity or shortness of breath. In patients with muscle disorders resultant alveolar hypoventilation may increase the risk of chronic respiratory failure during pregnancy [21,24]. It has also been shown that chronic maternal hypoxia is associated with decreased live birth rates [25].

An important risk relevant to women with respiratory disease is one where maternal hypoxia results in oxygen saturations less than 85%, and the livebirth rate is only 12%. While complication rates in pregnancy or childbirth were not higher in study subjects, compared to the CDC’s National Vital Statistics Report, there was an increase in the rate of stillbirths (3.8%) for the study subjects as compared to the national average (0.2–0.7%). However, these still births were found to be unrelated to Pompe disease. Complications reported in other studies related to bleeding risk were not observed in this group [26].

The distinguishing feature of Pompe disease among other disorders affecting the muscle is the existence of a disease specific treatment through Enzyme Replacement Therapy (ERT). Recombinant human alglucosidase alfa is the only FDA approved ERT for LOPD. In a randomized, placebo-controlled 18-month trial, alglucosidase alfa was shown to improve walking distance in the 6-min walk test and stabilization of pulmonary function. At 78 weeks, there was an absolute increase in Forced Vital Capacity (FVC); *p* = 0.006 [27].

In this study, two patients conceived while receiving alglucosidase alfa and one patient was started on ERT during pregnancy. There were no reports of adverse effects of ERT use in the four fetuses exposed during pregnancy. This cohort also demonstrates that patients and/or clinicians may have a preference to withhold therapy during a pregnancy. This decision could be based on the scarcity or the lack of data on the effects of alglucosidase alfa on the developing fetus. One other concern for the use of ERT during pregnancy could be the potential drug related immune hypersensitivity reactions. Infusion-related reactions, including anaphylaxis, urticaria, flushing, hyperhidrosis, chest discomfort, vomiting, and increased blood pressure were observed in up to 51% of the patients treated with alglucosidase alfa. In addition, systemic immune-mediated reactions have been observed with alglucosidase alfa such as skin involvement with deposition of anti-rhGAA antibodies, and severe inflammatory arthropathy in association with pyrexia and elevated erythrocyte sedimentation rate. Nephrotic syndrome secondary to membranous glomerulonephritis was observed in some Pompe disease patients treated with alglucosidase alfa who had persistently positive anti-rhGAA IgG antibody titers [10]. In the UK cohort, in four patients with LOPD with reported pregnancies, stopping ERT during pregnancy was associated with deterioration of Pompe disease related symptoms and emergence of infusion-related hypersensitivity reactions when ERT was resumed. Other human recombinant ERT products are used commonly during pregnancy in other Lysosomal Storage Disorders, such as Gaucher disease [28]. As discussed, LOPD is a progressive disorder, if untreated there is an expected decline both in peripheral muscle strength and respiratory function. Thus, the decision about ERT should be based on the individual patient, and special attention should be paid to already existing respiratory compromise, and prior history of immunogenicity against human recombinant alglucosidase alfa [29].

Limitations of the study include a small overall sample size as well as those receiving ERT prior to or during pregnancy and the potential bias of respondents being only women who have been pregnant. Other shortcomings of the study may include the retrospective study design and data collected by patient recall. Some of the concerns raised in the German cohort by Karabul et al., such as the presence of other co-existent conditions, could also hold valid for this study [30]. It is obvious that further data is needed to better analyze the effects of ERT on pregnancy outcomes and the effects of pregnancy on Pompe disease progression. Since Pompe disease has been now included into many states’ newborn screening programs, leading to more frequent and earlier diagnosis, prospective studies with more subjects and adequate control may be possible in future. However, the incidence (1 in 57,000 live births) of LOPD [31] may preclude an imminent large prospective multicenter trial to assess the effects of alglucosidase alfa and pregnancy in female patients with Pompe disease, and small studies such as this one may aid to develop practice guidelines.

## 4. Materials and Methods

In this observational case review study, 25 female subjects with a confirmed diagnosis (by mutation analysis, enzymatic assay, or both) of Pompe disease, age 18 years and over who were pregnant or were contemplating pregnancy, were enrolled at two centers under IRB-approved protocols (NCT01556516). On patient attestation, Pompe disease diagnosis was confirmed through the review of medical records.

After enrollment, patients were asked to complete a survey or were interviewed in person by the investigator(s). The questionnaire assessed the following: decision to become pregnant, fertility problems, severity of Pompe disease, functional limitations before and at the time of pregnancy, exposure to alglucosidase alfa during pregnancy, exposure to other known teratogens (alcohol, recreational drugs, etc.), pregnancy outcome, type of delivery, postpartum complications, worsening of muscle weakness; respiratory function during pregnancy, and issues with caring for the newborn. Medical records pertaining to the subject’s previous pregnancy(s), management of Pompe disease symptoms, and details of therapy were also reviewed. To prevent an ascertainment bias on positive results, during the consenting and interview process, it was emphasized that the information on the number of pregnancies without complications is as important as the information on the pregnancy with complications, by the investigator. All collected data was presented in the form of individual by-patient listings, and the confidentiality was maintained by providing an ID number to each patient enrolled in the study.

Reproductive history and pregnancy outcomes were analyzed in comparison to the general healthy population and with adult onset forms of muscular dystrophy. Findings for patients with Pompe disease will be compared with national data from the Centers for Disease Control and Prevention (CDC) for the general population. Continuous variables were summarized using descriptive statistics (n, mean, median, standard deviation, minimum and maximum). Categorical variables will be summarized using frequencies and percentages. Statistical analyses were performed using the SAS^®^ statistical software system (Statistic Analysis System, version 9.4, Cary, NC, USA).

## Figures and Tables

**Figure 1 life-10-00194-f001:**
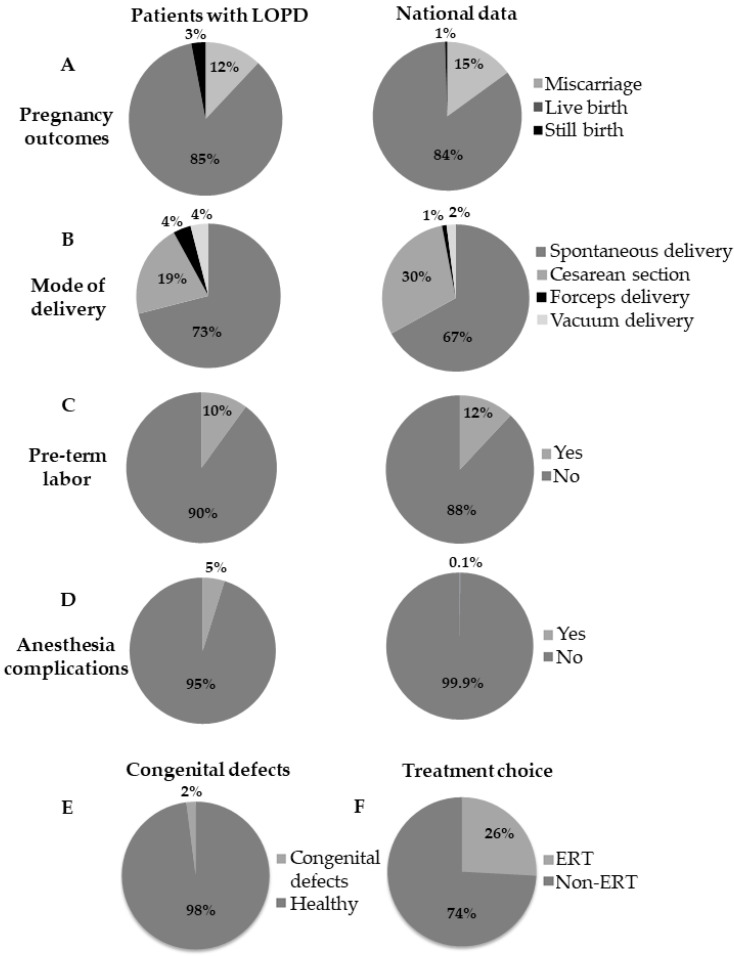
Pregnancy outcomes in LOPD: The frequency of pregnancy related events in 25 female patients with LOPD in comparison to the national data. This includes: (**A**) pregnancy outcomes; (**B**) mode of delivery; (**C**) pre-term labor; (**D**) anesthesia complications during delivery; along with (**E**) examining frequency of congenital defects in the study cohort; and (**F**) examining treatment decisions in pregnant patient with LOPD.

**Table 1 life-10-00194-t001:** Demographic data, Clinical history, diagnosis, and disease progression in female patients with LOPD.

Subject	Ethnicity	Age of Onset (Years)	Age at Diagnosis (Years)	Method of Diagnosis	Diagnosis Prior, During, or after Pregnancy	Presentation	Assistive Device
1	Other	7	33	Enzyme assay, molecular testing	After all pregnancies	Muscle cramps/muscle pain at end of the day	No
2	Ashkenazi Jewish	45	50	Muscle biopsy	After all pregnancies	Muscle pain, shortness of breath, limited exercise ability	No
3	Other	33	48	Enzyme assay	After all pregnancies	Muscle atrophy, right thigh	Cane (41 y.o.), scooter/walker (47 y.o.)
4	Northern European	20	48	Enzyme assay	After all pregnancies	Muscle weakness	Cane, walker, scooter (48 y.o.)
5	Northern European	44	44	Muscle biopsy, molecular testing	After all pregnancies	Proximal muscle weakness	Cane, walker, service dog (49 y.o.)
6	Northern European	18	24	Muscle biopsy, molecular testing	Before pregnancies	Difficulty going up stairs	AFOs
7	Northern European	41	46	Muscle biopsy	After all pregnancies	Leg weakness, lower back pain	Cane, walker
8	Other	27	32	Muscle biopsy	During pregnancy #4	Inability to squat and get up	Cane, scooter
9	N/A	20	36	Muscle biopsy, molecular testing	During pregnancy #4	Bruising and muscle soreness after Caesarean section	No
10	N/A	40–45	53	Muscle biopsy, molecular testing	After all pregnancies	Muscle weakness, hip pain	Wheelchair
11	N/A	30	46	Muscle biopsy, molecular testing	After all pregnancies	Shortness of breath	Cane, scooter
12	N/A	38	40	Muscle biopsy, molecular testing	After all pregnancies	Malaise, muscle weakness	No
13	N/A	51	55	Muscle biopsy, molecular testing	After all pregnancies	Leg pain, muscle weakness	Cane
14	N/A	30	35	Muscle biopsy, molecular testing	After all pregnancies	Malaise, muscle pain	Cane, walker
15	N/A	34	39	Muscle biopsy, molecular testing	After all pregnancies	Muscle weakness	Cane
16	Northern European	20	29	Muscle biopsy, molecular testing	N/A	Fatigue, elevated liver enzymes	No
17	Northern European	48	60	Enzyme assay, molecular testing	After all pregnancies	Fatigue, muscle weakness, elevated CK	No
18	N/A	30	42	Muscle biopsy, enzyme assay, molecular testing	After all pregnancies	Difficulty getting up from the floor	Cane
19	N/A	35	52	Muscle biopsy, enzyme assay, molecular testing	After all pregnancies	Fatigue	Cane
20	N/A	33	42	Muscle biopsy, enzyme assay, molecular testing	N/A	Difficulty climbing stairs	No
21	N/A	66	N/A	Muscle biopsy, enzyme assay, molecular testing	After all pregnancies	Shortness of breath, difficulty climbing stairs	No
22	N/A	31	31	Liver biopsy, molecular testing	After all pregnancies	Elevated liver enzymes	No
23	N/A	40	40	Muscle biopsy	After all pregnancies	Back pain, exercise intolerance	No
24	N/A	N/A	40	Muscle biopsy	After all pregnancies	Family history	Cane
25	N/A	45	49	Muscle biopsy	After all pregnancies	Muscle weakness	No

LOPD: late-onset Pompe disease, ERT: enzyme replacement therapy, AFO: ankle foot orthosis, N/A: not applicable, CK: creatine kinase, y.o.: years old.

**Table 2 life-10-00194-t002:** Enzyme Replacement Therapy (ERT) in female patients with LOPD.

Subject #	# of Pregnancies	# of Births	ERT Status
1	2	2	After all pregnancies
2	3	3	After all pregnancies
3	1	1	After pregnancy
4	4	4	After all pregnancies
5	1	1	After pregnancy 1
6	2 ^¥^	2(twins)	Before, during, and after pregnancy 2
7	2	2	After all pregnancies
8	6 ^¥^	5	Before, during and after pregnancy 6
9	6 ^¥¥^	4	During pregnancy 4
10	4 ^¥^	3	No ERT
11	2	2	No ERT
12	2	2	After all pregnancies
13	5	4 + 1 *	After all pregnancies
14	2 ^¥^	1 *	After all pregnancies
15	3	3	After all pregnancies
16	0	0	After cryopreservation of embryos
17	1	1	No ERT
18	2	2	After all pregnancies
19	2	2	After all pregnancies
20	0	0	After all pregnancies
21	2	2	After all pregnancies
22	3 ^¥^	2	No ERT
23	1	1	After all pregnancies
24	1	1	After all pregnancies
25	1	1	After all pregnancies

LOPD: late-onset Pompe disease, ERT: enzyme replacement therapy, * Stillbirth, ^¥^ Miscarriage.

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
