# Peer review of "Pregnancy Outcomes in Late Onset Pompe Disease"

_life, 2020, doi:10.3390/life10090194_

Round 1
Reviewer 1 Report
The authors present a cohort of woman with Pompe disease and prior pregnancies. A number of questions and concerns are raised.
The method section is at the end of the piece and not prior to the results. That might be the journal style but it raised initial questions that were answered at the end. The data is gathered from patient recollections and targeted record review. Because numerical data is analyzed, it would be instructive to have a general comment on how much data was recalled and how much was documented. I understand that precise information would be cumbersome to acquire. The issue is of particular interest for subjective strength worsening or improvement during and after pregnancy and delivery complication rates and conditions.
The title is a bit misleading. It suggests that the focus of the project is to determine the effect of ERT on pregnancy. Only a few in the cohort were treated or started on ERT during pregnancy. Perhaps the title should simply be: “Pregnancy Outcomes in Late Onset Pompe Disease”.
The case reports described individual pregnancy experiences are cited. One possible instructive one that notes issues discussed is: Zagnoli F, Leblanc A, Blanchard C. Pregnancy during enzyme replacement therapy for late-onset acid maltase deficiency. Neuromuscul Disord. 2013;23(2):180-181. doi:10.1016/j.nmd.2012.11.006.
I believe Lumizyme was categorized by the FDA as category B initially but changed to C and Myozyme as Category B based on mice and rabbit studies as you discuss. However, the FDA retired the letter system in favor of descriptors several years ago.
As for worsening during pregnancy, how did you separate declining from increased weight burden of pregnancy compared to disease progression? The methods note change in assistive devices will be a criterion.
Table 1 includes a column for assistive devices. It seems to be baseline requirements at the time of assessment but could be inferred that it relates somehow to pregnancy. That point should be clearer.
There is considerable general subject review in the introduction and discussion. I defer to the editor if this aspect should be better focused to the study goals.
Reviewer 2 Report
I like the topic since it's important to know about how Pompe affects pregnancy as a guide to women with LOPD who plan to become pregnant or to obgyn/MFM specialists. It's important to know that overall complication rate is not increased to overcome anxiety of patients and providers. It's also important to know that ERT is safe. However, the sample is small and there's a substantial recall bias since most patients were diagnosed postnatally. Coexisting conditions and age also could affect the muscle function. These limitations the authors state which is important. More subjects and randomization as well as prospective design would be preferable but it's a rare disease and it's difficult to do these kinds of studies. Yet, it's important to have a general idea about LOPD effect on pregnancy and as long as the limitations are stated (as done by the authors), I think this paper would be useful for the audience of LOPD patients and their families, reproductive planning and obgyn/MFM care. With more Pompe (both IOPD and LOPD) diagnosed by many states' newborn screen programs, prospective studies with more subjects and adequate control would be needed and maybe the authors could state that at the end.
